# Phosphatase of Regenerating Liver-1 (PRL-1)-Overexpressing Placenta-Derived Mesenchymal Stem Cells Enhance Antioxidant Effects via Peroxiredoxin 3 in TAA-Injured Rat Livers

**DOI:** 10.3390/antiox12010046

**Published:** 2022-12-26

**Authors:** Hee Jung Park, Ji Hye Jun, Jae Yeon Kim, Hye Jung Jang, Ja-Yun Lim, Si Hyun Bae, Gi Jin Kim

**Affiliations:** 1Department of Bio Convergence Science, CHA University, Seongnam-si 13488, Republic of Korea; 2Research Institute of Placenta Science, CHA University, Seongnam-si 13488, Republic of Korea; 3Department of Integrated Biomedical and Life Sciences, College of Health Science, Korea University, Seoul 03722, Republic of Korea; 4Department of Internal Medicine, Catholic University Medical College, Seoul 03312, Republic of Korea

**Keywords:** Liver disease, DNA damage and repair, stem cell therapy, placenta-derived mesenchymal stem cells, phosphatase-regenerating liver-1, antioxidant, peroxiredoxin 3

## Abstract

DNA damage repair is induced by several factors and is critical for cell survival, and many cellular DNA damage repair mechanisms are closely linked. Antioxidant enzymes that control cytokine-induced peroxide levels, such as peroxiredoxins (Prxs) and catalase (CAT), are involved in DNA repair systems. We previously demonstrated that placenta-derived mesenchymal stem cells (PD-MSCs) that overexpress PRL-1 (PRL-1(+)) promote liver regeneration via antioxidant effects in TAA-injured livers. However, the efficacy of these cells in regeneration and the role of Prxs in their DNA repair system have not been reported. Therefore, our objective was to analyze the Prx-based DNA repair mechanism in naïve or PRL-1(+)-transplanted TAA-injured rat livers. Apoptotic cell numbers were significantly decreased in the PRL-1(+) transplantation group versus the nontransplantation (NTx) group (*p* < 0.05). The expression of antioxidant markers was significantly increased in PRL-1(+) cells compared to NTx cells (*p* < 0.05). MitoSOX and Prx3 demonstrated a significant negative correlation coefficient (*R*^2^ = −0.8123). Furthermore, DNA damage marker levels were significantly decreased in PRL-1(+) cells compared to NTx cells (*p* < 0.05). In conclusion, increased Prx3 levels in PRL-1(+) cells result in an effective antioxidant effect in TAA-injured liver disease, and Prx3 is also involved in repairing damaged DNA.

## 1. Introduction

DNA damage is induced by UV rays, cancers, and several chemicals, such as thioacetamide (TAA) and carbon tetrachloride (CCl_4_) [1,2]. The liver detoxifies chemical substances by metabolic processes and inhibits radical oxygen species (ROS), preventing damage. Increased ROS resulting from oxidative stress induce an imbalance of oxidation and reduction in the mitochondria, alter fatty acid oxidation (FAO), cause mitochondrial DNA damage, and can lead to abnormal metabolic conditions. Hepatocytes with DNA damage are distorted and dysfunctional and accumulate abnormal lipid droplets [3,4,5]. Additionally, it has been reported that lipotoxicity caused by excessive lipid accumulation in cells causes both mitochondrial and DNA damage [6,7,8].

DNA damage is repaired via four pathways: base excision repair (BER), classical nonhomologous end-joining (NHEJ), alternative NHEJ, and homologous recombination (HR). BER removes and replaces damaged DNA through DNA glycosylation. Classical and alternative NHEJ repair double-strand breaks. HR repairs double-or single-stranded DNA damage by exchanging nucleic acids with those of a similar or identical molecular strand [9,10,11].

In addition to these four paths, it has been reported that DNA can be repaired through thiol-based antioxidant mechanisms. Antioxidant enzymes that control cytokine-induced peroxide levels, such as peroxiredoxins (Prxs), superoxide dismutase (SOD), and catalase (CAT), are involved in the thiol system [12,13].

There are three types of Prxs: typical 2-Cys Prxs, atypical 2-Cys Prxs, and 1-Cys Prxs [14]. Prx1–6s are present in mammalian cells. Interestingly, Prx3 is a representative antioxidants enzyme that is mitochondria-specific. Prx3 belongs to the typical 2-Cys Prx group and is catalyzed by the 2-Cys residue. The peroxide is oxidized to sulfonic acid (Cys-SOH) by the cysteine residue. After oxidation, Prx3 is reduced to an active thiol by thioredoxin/thioredoxin [15,16]. Additionally, Prx3 suppresses steroidogenesis, apoptosis, and autophagy by reducing hyperoxides, such as H_2_O_2_ [17,18]. Additionally, Prx3 suppresses adipogenesis in alcoholic liver disease by removing ROS and regulating PTEN oxidation [19].

We previously demonstrated that placenta-derived mesenchymal stem cells (PD-MSCs) have antioxidant and regenerative effects in TAA-injured rat livers and ovaries [20]. In addition, it was confirmed that ovarian function and follicle development were improved through antioxidant effects by the regulation of heme oxygenase-1 (HO-1), SOD, and CAT in an ovariectomized rat model after naïve PD-MSC transplantation. Naïve PD-MSCs also inhibit lipid accumulation and the activation of adipogenic factors, such as adiponectin, peroxisome proliferator-activated receptor gamma (PPARγ), and CCAAT/enhancer-binding protein alpha (C/EBPα) in primary orbital fibroblasts from Graves’ ophthalmic disease patients [21].

The phosphatase of regenerating liver-1 (PRL-1, PTP4A1) gene product is involved in hepatic regeneration and lipid metabolism [22,23]. We overexpressed PRL-1 in naïve PD-MSCs to maximize their effect (PRL-1(+) cells) and confirmed that these cells promote liver regeneration by improving ATP production and mitochondrial dynamics in a chronic cholestatic liver cirrhosis rat model [24]. However, the role of Prx3 in the antioxidant system and the efficacy of PRL-1(+) in improving DNA repair has not been reported. Therefore, our objectives were to analyze the ability of PRL-1(+) cells to improve DNA repair via Prx3 and to evaluate the correlation between PRL-1(+) cells and the Prx3 intramitochondrial-based DNA repair system.

## 2. Materials and Methods

### 2.1. Animals

Seven-week-old female Sprague–Dawley rats were obtained (Orient Bio Inc., Seongnam, Republic of Korea) and maintained in an air-conditioned facility. Hepatic failure was induced by intraperitoneal injection with Thioacetamide (TAA, 300 mg/kg, twice a week and 12 weeks) (Sigma-Aldrich, St. Louis, MO, USA). The rats were assigned to one of the following groups: normal (Nor; *n* = 4), TAA-injured nontransplantation (NTx; *N* = 9), naïve PD-MSC transplantation (naïve; *n* = 8), and PRL-1(+)-overexpressing naïve transplantation (PRL-1(+); *n* = 7). Intravenous transplantation of naïve and PRL-1(+) cells (2 × 10^6^) was performed at 8 weeks. The rats were sacrificed at 12 weeks. The experimental protocol was approved by the Institutional Animal Care and Use Committee of CHA University, Seongnam, Republic of Korea (IACUC-190048).

### 2.2. Cell Culture

WB-F344s, immortalized rat hepatocyte-like epithelial cells, were maintained in alpha-modified minimal essential medium (α-MEM; Clontech, Kusatsu, Japan) supplemented with 10% fetal bovine serum (FBS; Gibco, BRL, Langley, OK, USA) and 1% penicillin/streptomycin (P/S; Gibco). Naïve PD-MSCs were isolated from the chorionic plate of human placental cells (IRB 07-18), and PRL-1-overexpressing PD-MSCs were maintained in alpha-modified minimal essential medium (α-MEM; Clontech) supplemented with 10% fetal bovine serum (FBS; Gibco), 1% penicillin/streptomycin (P/S; Gibco), fibroblast growth factor 4 (FGF4; Peprotech, Cranbury, NJ, USA) (25 ng/mL), and heparin (Sigma-Aldrich) (1 μL/mL). The cells were maintained in <5% CO_2_ at 37 °C.

### 2.3. Blood Chemistry

Aspartate aminotransferase (AST), alanine aminotransferase (ALT), total bilirubin and albumin (ALB), interleukin-6 (IL-6), and tumor necrosis factor-alpha (TNF-α) levels in the serum samples were evaluated by the Southeast Medi-Chem Institute (Busan, Republic of Korea).

### 2.4. Quantitative Reverse Transcription–Polymerase Chain Reaction (qRT–PCR)

Total RNA was extracted from liver samples using TRIzol reagent (Invitrogen, Carlsbad, CA, USA), chloroform (Sigma-Aldrich) and isopropanol (Merck, Darmstadt, Germany). RNA was reverse transcribed into cDNA using Superscript III (Invitrogen) and RNaseout (Invitrogen). cDNA was amplified with designed primers and detected using SYBR green master mix (Roche, Diagnostics, Besal, Switzerland). Rat GAPDH was used as an internal control. All experiments were performed in triplicate. The primer sequences are shown in Table 1.

### 2.5. Western Blotting

Whole liver tissues were homogenized with LN_2_ and lysed on ice with RIPA buffer (SIGMA-Aldrich) containing protease inhibitor cocktail (Roche) and phosphate inhibitor cocktail II (A.G Scientific, Inc., San Diego, CA, USA). Protein lysates were separated by sodium dodecyl sulfate–polyacrylamide gel electrophoresis (SDS–PAGE) and transferred to polyvinylidene difluoride membranes (PVDF; Bio-Rad, Hercules, CA, USA). Membranes was blocked with 5% BSA (RDT, Eight Mile Plains, Australia) for 1 h at RT and then incubated with primary antibodies overnight at 4 °C. Antibodies against Ku70 (1:1000; Thermo Fisher Scientific, Waltham, MA, USA), heme oxygenase-1 (HO-1; 1:1000; Novus, Berkley, MI, USA), superoxide dismutase (SOD; 1:1000; Cell Signaling, Dallas, TX, USA), peroxiredoxin 1 (Prx1; 1:1000; Abcam, Cambridge, UK), peroxiredoxin 2 (Prx2; 1:1000; Abcam), peroxiredoxin 3 (Prx3; 1:1000; Abcam), peroxisome proliferator-activated receptor gamma coactivator 1 alpha (PGC1α; 1:1000; Novus), nuclear factor erythroid 2-related factor 2 (Nrf2; 1:1000; BIOSS antibodies, Woburn, MA, USA), alpha-smooth muscle actin (α-SMA; 1:1000; DAKO, Santa Clara, CA, USA), and albumin (ALB; 1:1000; Novus) were used. All experiments were performed in duplicate or triplicate. The intensity of each band was quantified by ImageJ software (NIH, Bethesda, MD, USA).

### 2.6. Enzyme-Linked Immunosorbent Assay (ELISA)

Protein concentrations were normalized with a bovine serum albumin protein assay kit, and proteins were processed with a total antioxidant assay (BIOMEX, Seoul, Republic of Korea), TBARS assay (BIOMEX), glutathione colorimetric detection (ARBOR assay, Ann Arbor, MI, USA) and LDH assay (lactate dehydrogenase; LDH; Abcam), according to the manufacturer’s instructions. Experiments were performed in triplicate.

### 2.7. Analysis of mtDNA Copy Number

A gDNA isolation kit was used for manual extraction (Bioneer, Daejeon, Republic of Korea). gDNA was extracted from homogenized TAA-injured rat liver to analyze the mtDNA copy number. qRT–PCR amplification was conducted with specific primers containing 250 ng of gDNA and nuclear DNA primers with 2X TaqMan Universal Master Mix (Roche), according to the manufacturer’s instructions. These experiments were performed in triplicate.

### 2.8. Histological Analysis

Liver tissues were fixed in 10% formalin and embedded in paraffin. The samples were sectioned at a thickness of 7 μm. The samples were stained with hematoxylin and eosin (H&E) (DAKO, Santa Clara, CA, USA), oil red O (LABORIMPEX, Bruxelles, Brussel) and Sirius red. All experiments were conducted in triplicate.

### 2.9. Immunohistochemistry

Tissues were fixed in 10% formalin in paraffin. The samples were sectioned at a thickness of 7 μm. Primary antibodies against proliferating cell nuclear antigen (PCNA, Santa Cruz Biotechnology, Dallas, TX, USA; 1:200) and 8-hydroxy-2′-deoxyguanosine (8-OHdG, Abcam; 1:200) were incubated with tissues at 4 °C overnight. The images were scanned by a digital histologic scanner (3DHISTECH Ltd., Budapest, Hungary). These experiments were performed in triplicate.

### 2.10. Immunofluorescence

Tissues were fixed in 10% formalin in OCT compound (Leica, Buffalo Grove, IL, USA) for the analysis of the localization of Prx3 in mitochondria. The samples were sectioned at a thickness of 7 μm. The tissues were blocked using blocking solution (DAKO) for 1 h in the dark. The primary antibody against Prx3 (1:100; Abcam) was added to the diluent solution (DAKO) and incubated with sections at 4 °C overnight. The secondary antibody, Alexa Fluor 488 goat anti-rabbit immunoglobulin IgG (Invitrogen), was then added for 1 h. The slides were counterstained with 4,6-diamidino-2-phenylindole (DAPI, Invitrogen). The images were observed with a confocal microscope (Zeiss, Oberkochen, Germany). These experiments were performed in triplicate.

### 2.11. TUNEL Assay

Rat liver tissues were fixed in 10% formalin in paraffin. The samples were sectioned at a thickness of 7 μm. Rat liver tissues were used for the TUNEL assay with an HRP-DAB TUNEL staining kit (Abcam). All stained slides were scanned.

### 2.12. Statistical Analysis

Statistical analyses were performed with Student’s *t* test or one-way ANOVA using the R program (R-project, Auckland, New Zealand) and Graph Prism 9 (GraphPad Software, San Diego, CA, USA). All experiments were repeated in duplicate to triplicate under the same conditions. A *p* value < 0.05 was considered statistically significant.

## 3. Results

### 3.1. Attenuation of Cell Death in Naïve and PRL-1(+) Cells Transplanted into TAA-Injured Rat Livers

To induce a cirrhosis model, rats received intraperitoneal injections of TAA (300 mg/kg) twice a week for 12 weeks [25]. The rats were assigned to one of the following groups: normal (Nor; *n* = 4), TAA-injured nontransplantation (NTx; *n* = 9), naïve PD-MSC transplantation (naïve; *n* = 8), and PRL-1(+)-overexpressing naïve transplantation (PRL-1(+); *n* = 7). Intravenous transplantation of naïve and PRL-1(+) cells (2 × 10^6^) was performed at 8 weeks. Inflammatory markers (IL-6 and TNF-α) were measured in the serum of naïve and PRL-1(+) cell-transplanted TAA-injured rats. As shown in Figure 1a,b, IL-6 and TNF-α levels were significantly increased in the NTx group relative to the naïve and PRL-1(+) groups (** p* < 0.05; Figure 1a,b). Additionally, histological changes related to liver injury and fibrosis were observed in liver tissues stained with hematoxylin and eosin (H&E) and subjected to terminal deoxynucleotidyl transferased UTP nick and labeling (TUNEL) assays. With a TUNEL assay, apoptosis can be identified by marking fragmented DNA ends; here, TUNEL assay results showed that apoptosis was reduced in the naïve and PRL-1(+) groups compared to the NTx group. Quantification of TUNEL staining in the NTx group showed a significant decrease in naïve and PRL-1(+) cells compared with the number of cells in the NTx group (** p* < 0.05; Figure 1c,d). Additionally, necrotic, and apoptotic markers (LDH, and caspase3) were assessed. LDH activity was significantly increased in the naïve and PRL-1(+) groups compared to the NTx group (** p* < 0.05; Figure 1f).

Lipid metabolism is correlated with increased apoptosis. Thus, lipid metabolism was analyzed to verify the above findings, and total cholesterol and HDL levels were significantly increased in naïve cells compared to the NTx cells (** p* < 0.05; Figure 1h,j). Leptin, LDL, PPARγ, adiponectin, adipsin, FABP4, and LPL in the naïve and PRL-1(+) groups were decreased relative to levels in the NTx group (** p* < 0.05; Figure 1i,k–p). Therefore, these data demonstrated that transplantation of naïve and PRL-1(+) cells enhances liver regeneration and protects against cell death in TAA-treated rat livers.

### 3.2. The Effect of Naïve and PRL-1(+) Cells Transplanted into TAA-Injured Rat Livers on DNA Repair

DNA damage induced by various hazardous substances can lead to liver damage. DNA damage has been reported to cause inflammation and worsen liver damage [26]. As shown in Figure 2a, the DNA damage marker 8-oxo-2′-deoxyguanosine (8-OHdG) was detected in DNA-damaged TAA-injured rat livers. 8-OHdG positivity was significantly increased in the NTx group compared with the naïve and PRL-1(+) groups (** p* < 0.05, Figure 2b). Additionally, DNA damage marker levels (γH2AX and Ku70) were significantly decreased in the naïve and PRL-1(+) groups relative to those in the NTx group (** p* < 0.05, Figure 2c,d). In particular, γH2AX levels were significantly decreased in the PRL-1(+) group compared to the naïve group. These results confirm that transplanted naïve and PRL-1(+) cells in TAA-injured rat livers facilitate DNA repair.

### 3.3. Regulation of Reactive Oxygen Species (ROS) in Naïve and PRL-1(+) Cells Transplanted into TAA-Injured Rat Livers

TAA causes cirrhosis of the liver by inducing oxidative stress. Sustained oxidative stress accelerates liver cirrhosis [20]. As shown in Figure 3a,b, the mitochondrial ROS levels in TAA-injured rat livers were significantly increased in the NTx group relative to those in the naïve and PRL-1(+) groups (** p* < 0.05, Figure 3a,b). Additionally, the expression of NAPDH oxidase 4 (Nox4) was significantly decreased in the naïve and PRL-1(+) groups. (** p* < 0.05, Figure 3c). The levels of the downstream factors of Nox4, such as heme oxygenase-1 (HO-1) and glutathione peroxidase (GPX), were significantly decreased in the naïve and PRL-1(+) groups and significantly increased compared with the PRL-1(+) group (** p* < 0.05, Figure 3d,e). When antioxidants were confirmed through ELISA at the serum level, total antioxidant levels were decreased in the NTx group relative to those in the naïve and PRL-1(+) groups.

Additionally, the PRL-1(+) group exhibited significantly increased total antioxidant levels relative to those in the naïve group (** p* < 0.05, Figure 3f). We analyzed the levels of thiobarbituric acid reactive substance (TBARS) by ELISA to assess lipid peroxidation in TAA-injured rat livers. TBARS was significantly increased in the NTx group compared with the naïve and PRL-1(+) groups (** p* < 0.05, Figure 3g). Thus, antioxidant marker glutathione levels were higher in the NTx group than those in the naïve and PRL-1(+) groups (Figure 3h). The protein level of HO-1 in the naïve group decreased slightly compared to that in the NTx group, but it was increased in the PRL-1(+) group (Figure 3i). Conversely, the expression of superoxide dismutase (SOD) was increased in the naïve group relative to that of the NTx group but was significantly decreased in the PRL-1(+) group (** p* < 0.05, Figure 3j). Finally, the expression of catalase (CAT), the end product of antioxidation, gradually increased in the naïve and PRL-1(+) groups relative to that of the NTx group (Figure 3k). In conclusion, the antioxidant effect improved gradually as the ROS level increased in TAA-injured rat livers in the naïve and PRL-1(+) groups.

### 3.4. Effect of the Mitochondrial Peroxiredoxin Family in Naïve-and PRL-1(+)-Cell-Transplanted, TAA-Injured Rat Livers

Oxidative stress is a biochemical reaction characterized by the excess production of free radicals and reactive metabolites that are potentially harmful to the organism. Antioxidant enzymes, such as peroxiredoxin, SOD, CAT and GPx reduce the increased ROS [27]. The protein levels of peroxiredoxin 1-3 were analyzed and were found to gradually and significantly increase in the naïve and PRL-1(+) groups compared to the NTx group (** p* < 0.05, Figure 4a–c). Hence, an association of peroxiredoxin with mitochondria was identified through immunofluorescence staining. It was confirmed that Prx3 protein levels gradually increased in the naïve and PRL-1(+) groups compared to those in the NTx group, but MitoSOX levels gradually decreased. Both protein levels indicated a negative correlation (*R*^2^ = −0.8123; ** p* < 0.05, Figure 4d–f). Therefore, the ROS increase after transplantation of naïve and PRL-1(+) cells into the livers of TAA-injured mice was confirmed to be alleviated by the antioxidant effect of peroxiredoxin 3 in mitochondria.

### 3.5. The Effect of Naïve and PRL-1(+) Cell Transplantation on Mitophagy in TAA-Injured Rat Livers

Mitophagy, known as macroautophagy, selects damaged mitochondria for degradation. The cell double-membrane autophagosome selectively targets the damaged region of the entire mitochondria, causing autophagosomes to fuse with ribosomes for the degradation of the damaged mitochondria [28]. Mitophagy markers, including PTEN-induced kinase 1 (PINK1) and E3 ubiquitin ligase (PARKIN), were analyzed to analyze mitophagy in naïve and PRL-1(+) cell-transplanted TAA-injured rat livers. The mRNA and protein levels of the markers were significantly increased in the naïve and PRL-1(+) groups compared to those in the NTx group. (** p* < 0.05, Figure 5a–d). PINK1 expression was confirmed to be localized in the mitochondria in the tissue cytoplasm, and the results were consistent between the mRNA and protein levels (** p* < 0.05, Figure 5e,f). Taken together, these results confirmed that mitophagy is activated through PINK1-PARKIN signaling via naïve and PRL-1(+) transplantation in TAA-treated rat livers.

### 3.6. Analysis of Mitochondrial Biogenesis Efficacy through Naïve and PRL-1(+) Transplantation in TAA-Injured Rat Livers

As shown in Figure 6a,b, mitochondrial biogenesis activity was confirmed through mtDNA copy number and ATP production assays. The mtDNA copy number was significantly increased in the naïve and PRL-1(+) groups relative to that of the NTx group (** p* < 0.05, Figure 6a). In addition, ATP production was slightly decreased in the naïve group compared to that in the NTx group; however, ATP production was significantly increased in the PRL-1(+) group (** p* < 0.05, Figure 6b). Additionally, mitochondrial fission and fusion markers (e.g., Fis1, Drp1, OPA1, and Mfn2) were significantly decreased in the naïve and PRL-1(+) groups compared with the NTx group (** p* < 0.05, Figure 6c–f). The levels of phosphoinositide 3-kinase class III (PI3K class III), an autophagy marker, were increased in the naïve group compared to those in the NTx group and were decreased significantly in the PRL-1(+) group compared to those in the naïve group (** p* < 0.05, Figure 6g). mTOR, a negative modulator of autophagy, was significantly decreased in the naïve and PRL-1(+) groups relative to that of the NTx group (** p* < 0.05, Figure 6h).

On the other hand, AMP-activated protein kinase-alpha (AMPKα) was significantly increased in the naïve and PRL-1(+) groups relative to that of the NTx group, and there was a significant difference between the naïve and PRL-1(+) groups (** p* < 0.05, Figure 6i). Additionally, cyclin D1, autophagy-related 7 (ATG 7), and microtubule-associated protein 1A/1B light chain 3B (LC3B) levels were analyzed. The expression levels were significantly increased in the naïve and PRL-1(+) groups relative to those of the NTx group (** p* < 0.05, Figure 6j–l). Therefore, it was confirmed that when naïve and PRL-1(+) cells were transplanted into the TAA-injured rat model, mitochondrial biogenesis and autophagy were positively regulated.

### 3.7. Expression of Antioxidant Factors in TAA-Treated WB-F344s

The antioxidant efficacy of naïve cells was confirmed inWB-F344s, which are immortalized rat hepatocyte-like epithelial cells. The ROS level in the mitochondria was significantly reduced in the cells cocultured with naïve and PRL-1(+) cells compared to the TAA-treated cells. However, there was a significant increase in the PRL-1 knockdown groups (** p* < 0.05, Figure 7a,b).

In addition, when peroxiredoxin 3 was evaluated, it was found to be increased in the naïve and PRL-1(+) cocultivated groups compared with the TAA-treated group and decreased in the PRL-1(+) cocultured group compared to the naïve cocultured group. On the other hand, in each naïve cocultured group, it was significantly decreased in the PRL-1 knockdown group, but it was increased in the PRL-1(+) cocultivated group (** p* < 0.05, Figure 7c,d). Next, the DNA repair markers8-OHdG and γH2AX were analyzed. Additionally, 8-OHdG and γH2AX were downregulated in the naïve and PRL-1(+) cocultured groups relative to the levels of each in the TAA-treated group (** p* < 0.05, Figure 7e,f). The expression of PINK and PARKIN was significantly increased in the naïve and PRL-1(+) cocultivated groups relative to that of the TAA-treated group. It was confirmed that the PRL-1 knockdown groups had significantly decreased expression (** p* < 0.05, Figure 7g,f). Therefore, it was confirmed that the coculture of naïve and PRL-1(+) cells with WB-F344s induces antioxidant effects.

### 3.8. The Effect of Naïve and PRL-1(+) Cell Transplantation on Cell Proliferation in TAA-Injured Rat Livers

Although the normal liver has excellent regenerative capacity, this capacity is lost during liver cirrhosis [29]. The levels of proliferating cell nuclear antigen (PCNA), a proliferation marker, were measured in TAA-injured rat livers by immunohistochemistry. PCNA expression was significantly increased in the naïve and PRL-1(+) groups compared to the levels in the NTx group (** p* < 0.05, Figure 8a,b).

Additionally, when the levels of albumin (ALB), a hepatic regeneration marker, were measured in the serum and at the protein level, they were found to be significantly increased in the naïve and PRL-1(+) groups compared to the NTx group (** p* < 0.05, Figure 8c,d). These results demonstrate that the transplantation of naïve and PRL-1(+) cells in a TAA-induced rat model enhances cellular proliferative capacity.

## 4. Discussion

Mitochondrial DNA (mtDNA) damage is induced by various endogenous and environmental factors [30]. It attenuates ROS in cells to reduce oxidative stress and increases mutations in mitochondrial DNA, resulting in efforts to repair the damaged mitochondrial DNA. According to a recent study, mtDNA damage in cardiovascular disease is repaired through the Pol gamma-Nrf2-cGAS-STING pathway by reducing mitochondrial ROS through the regulation of oxidative stress [31,32].

Oxidative enzymes, such as nicotinamide adenine dinucleotide phosphate (NAPDH), are involved in regulating mitochondrial ROS, and enzymes such as thioredoxin, peroxiredoxin, glutathione, and SOD induce oxidative stress through the redox process [33,34]. Accordingly, our MitoSOX assay results confirmed that TAA increased mitochondrial ROS [35,36]. Increased mitochondrial ROS activate the oxidation/reduction factor peroxiredoxin; PD-MSCs are known to have antioxidant functions, and PD-MSCs overexpressing PRL-1 were assessed. These cells attenuated the ROS increase resulting from peroxiredoxin expression, which also affected the mitochondria (Figure 2).

In addition, it has been reported that increased lipotoxicity increases oxidative stress and induces DNA damage in cells [37]. Chao and colleagues also reported that DNA damage in the liver reduces the regenerative capacity of the liver and induces apoptosis [32]. We hypothesized that TAA administration induced DNA damage in rat liver hepatocytes and that naïve and PRL-1(+) cell transplantation could inhibit mitochondrial DNA damage through a repair system triggered via Prx3-induced antioxidant effects. In addition, in a previous study, uracil-DNA glycosylase (UNG1), a protein involved in nucleotide cut repair in mitochondria, was found to protect against increased ROS and prevent mtDNA damage through the interaction between UNG1 and Prx3 [38]. Consistently, we confirmed that when PD-MSCs and PRL-1-overexpressing PD-MSCs were transplanted, the increased peroxiredoxin expression attenuated the increased ROS levels and prevented the damage to mtDNA (Figure 4).

In addition, the disadvantages of mesenchymal stem cells were alleviated by using mesenchymal stem cells overexpressing PRL-1, which provided the cells with enhanced functions. According to Kim et al. their collogues, PD-MSCs with enhanced PRL-1-mediated function promote liver function regeneration, increase mtDNA protection from damage and increase ATP production [23]. In addition, PD-MSCs with enhanced PRL-1 function improve ER stress, vascular remodeling, follicular development, and liver function. The therapeutic efficacy of PD-MSCs is basically caused by the transplanted PD-MSCs into damaged liver tissue. Since PD-MSCs engrafted in damaged liver tissue after transplantation secretes various effective factors through “cross-talk” with surrounding tissues, colonization of engrafted PD-MSCs into damaged tissue is one of the important actors in confirming therapeutic efficacy. Therefore, in previous reports, in order to identify donor cells transplanted into liver tissue, each of the human-specific nuclear and hepatocyte differentiation-related marker antibodies were demonstrated by the IHC method and fluorescence microscopic analysis of PKH-67 labeled PD-MSCs was performed [39]. Accordingly, in this study, it was confirmed that hepatic regeneration was improved by transplanting PRL-1-enhanced placenta-derived stem cells into the TAA-induced model and the BDL model.

Specifically, in the case of mitochondrial damage, it has been reported that damaged DNA is repaired by oxidation through Prx3 [31]. Prx3 has also been shown to act as an antioxidant by inhibiting an increase in ROS in mitochondria [34]. According to a recent study, in intrahepatic cholestasis (ICP) in pregnant women, bile acids increase through the fetal and maternal circulation and induce oxidative stress in the fetus, thereby inducing dysfunction in mitochondria. Prx3 is a p38-mitogen-activated protein that has been shown to reduce growth arrest and cellular senescence by inhibiting the induction of MAP kinase (MAPK), p21 WAF1/CIP and p16INK4A [40]. Consistently, our experimental results confirmed that naïve and PRL-1(+) cell transplantation regulated TAA-induced ROS increase, reducing oxidative stress (Figure 3).

Therefore, in this study, the expression of peroxiredoxin was assessed and confirmed based on the hypothesis that peroxiredoxin family proteins influence the Srx-Trx mechanism [41]. Our results showed that all of the stem cell transplantation groups gradually increased compared to the nontransplantation group, and it was thought that stem cell transplantation would further enhance antioxidant activity. According to previous studies, peroxiredoxin reduces peroxide levels and inhibits adipogenesis in mitochondria by decreasing the expression of mitochondrial adipogenic and ROS-producing genes [42]. In a previous study, it was confirmed that the increase in ROS levels was significantly decreased in the TAA group, and the DNA markers 8-OHdG, γH2AX, and Ku70 were analyzed to determine whether DNA damage was induced. Placenta-derived mesenchymal stem cells and placental stem cells with enhanced PRL-1 function were investigated. The results indicate that ROS level increases resulting from peroxiredoxin activity were decreased by placenta-derived mesenchymal stem cells and mesenchymal stem cells with enhanced PRL-1 function (Figure 2).

## 5. Conclusions

In conclusion, PRL-1-overexpressing PD-MSC transplantation is a potential new treatment strategy for chronic liver disease that regulates ROS through the Prx3-based DNA repair system.

## Figures and Tables

**Figure 1 antioxidants-12-00046-f001:**
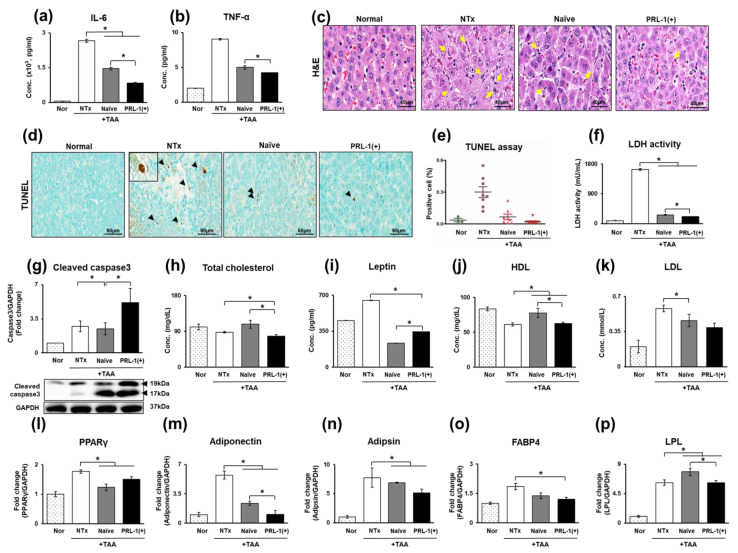
Effect of naïve and PRL-1(+) cells on apoptosis in TAA-injured rat livers. The expression of inflammation and apoptosis markers in TAA-injured rat livers (**a**) IL-6, (**b**) TNF-α, (**c**) H&E; yellow arrows: apoptotic cells, (**d**,**e**) TUNEL assay, (**f**) LDH activity, (**g**) cleaved caspase 3. The expression of lipid metabolism markers in the TAA-injured rat model in the serum and via mRNA analysis (**h**) total cholesterol, (**i**) leptin, (**j**) HDL, (**k**) LDL, (**l**) PPARγ, (**m**) adiponectin, (**n**) adipsin, (**o**) FABP4, and (**p**) LPL. All experiments were repeated in duplicate to triplicate. * *p* < 0.05.

**Figure 2 antioxidants-12-00046-f002:**
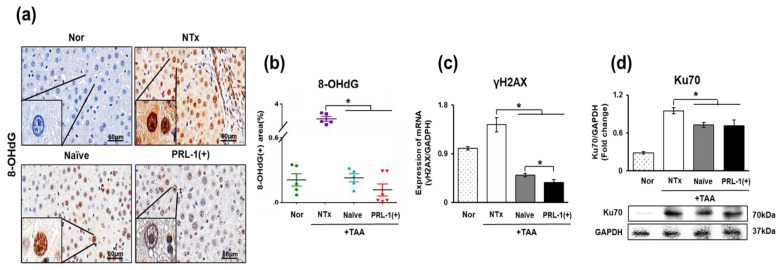
Effects of naïve and PRL-1(+) DNA repair systems in TAA-injured rat livers. The expression of DNA damage and repair markers in TAA-injured rat livers by IHC, mRNA level and protein level. (**a**,**b**) 8-OHdG, (**c**) γH2AX, and (**d**) Ku70. All experiments were repeated in duplicate to triplicate. * *p* < 0.05.

**Figure 3 antioxidants-12-00046-f003:**
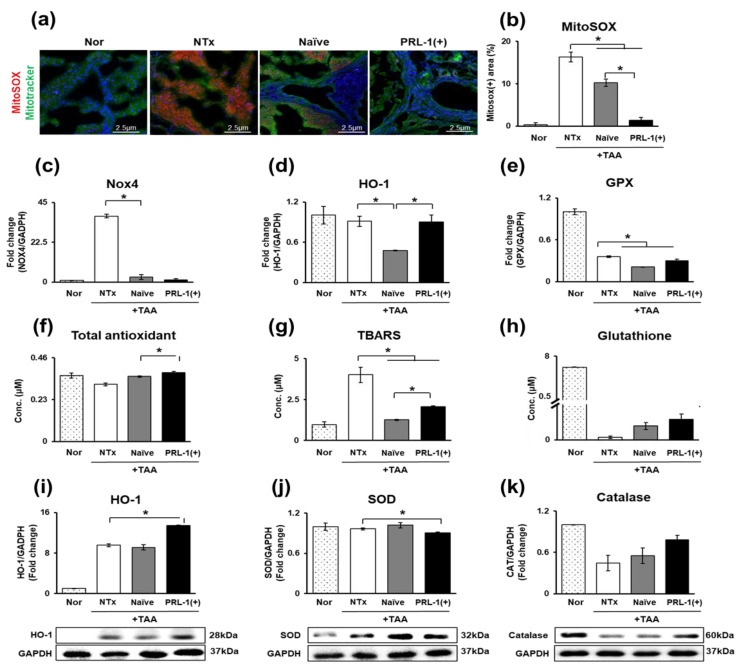
Expression of ROS in naïve and PRL-1(+) transplanted rat livers. The expression of MitoSOX in TAA-injured rat liver (**a**) MitoSOX and Mitotracker, (**b**) MitoSOX intensity. The expression of ROS in TAA-injured rat liver by qRT–PCR, ELISA and Western blot (**c**) Nox4, (**d**) HO-1, (**e**) GPx, (**f**) Total antioxidant, (**g**) TBARS, (**h**) Glutathione, (**i**) HO-1, (**j**) SOD, (**k**) CAT. All experiments were repeated in duplicate to triplicate. * *p* < 0.05.

**Figure 4 antioxidants-12-00046-f004:**
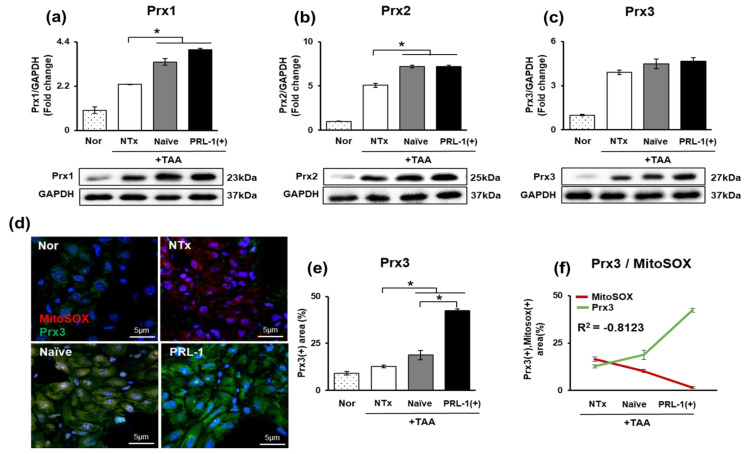
Peroxiredoxin family in naïve and PRL-1(+) cell-transplanted rat livers. The expression of the peroxiredoxin family in TAA-injured rat liver by Western blot and IF (**a**) Prx1, (**b**) Prx2, (**c**) Prx3, (**d**) MitoSOX and Prx3, (**e**) Prx3 intensity, (**f**) Prx3 and MitoSOX graph. All experiments were repeated in duplicate to triplicate. * *p* < 0.05.

**Figure 5 antioxidants-12-00046-f005:**
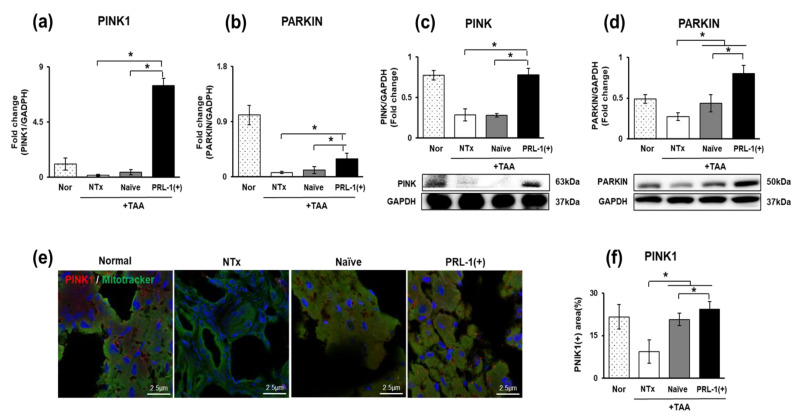
Regulation of mitophagy in TAA-injured rat livers. The expression of mitophagy in TAA-injured rat livers measured by qRT–PCR and Western blot (**a**,**c**) PINK1, (**b**,**d**) PARKIN. Localization of PINK1 in TAA-injured rat livers demonstrated by IF (**e**,**f**). All experiments were repeated in duplicate to triplicate. ** p* < 0.05.

**Figure 6 antioxidants-12-00046-f006:**
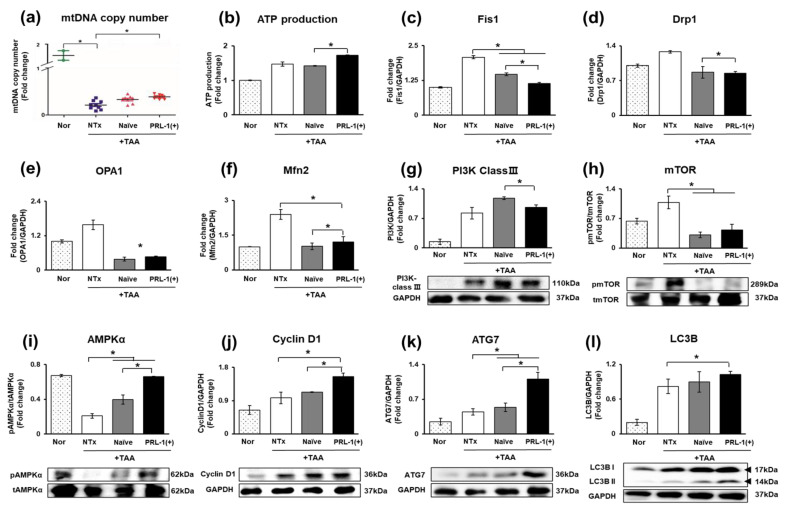
Autophagy in TAA-injured rat liver. The expression of mitochondrial homeostasis in TAA-injured rat liver by gDNA and ELISA. (**a**) mtDNA copy number, (**b**) ATP production. The expression of mitochondrial fission and fusion markers in TAA-injured rat liver by qRT–PCR (**c**) Fis1, (**d**) Drp1, (**e**) OPA1, (**f**) mfn2. The expression of autophagy markers in TAA-injured rat liver by Western blot (**g**) PI3K class Ⅲ, (**h**) mTOR, (**i**) AMPKα, (**j**) Cyclin D1, (**k**) ATG7, (**l**) LC3B.All experiments were repeated in duplicate to triplicate. * *p* < 0.05.

**Figure 7 antioxidants-12-00046-f007:**
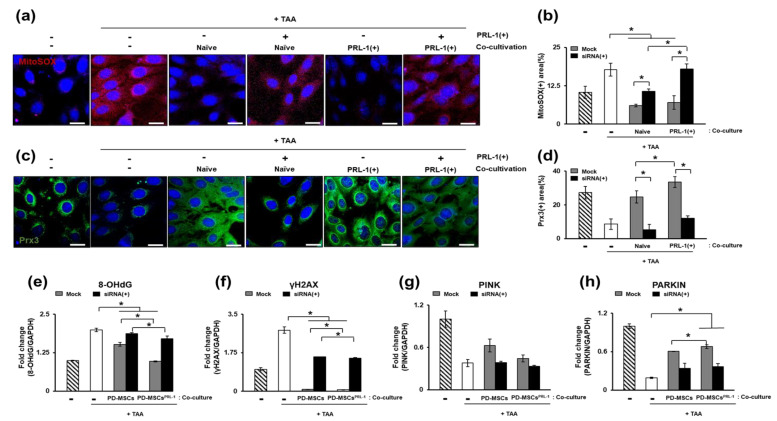
Expression of antioxidants in TAA-treated WB-F344s. The expression of antioxidant markers in TAA-treated WB-F344s (**a**,**b**) MitoSOX, (**c**,**d**) Prx3, (**e**) 8-OHdG, (**f**) γH2AX, (**g**) PINK, and (**h**) PARKIN. All experiments were repeated in duplicate to triplicate. * *p* < 0.05.

**Figure 8 antioxidants-12-00046-f008:**
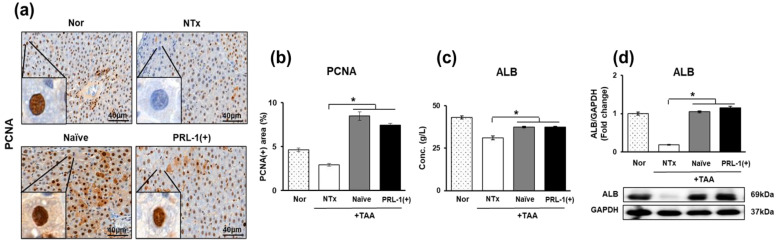
Effects of naïve and PRL-1(+) cells on proliferation in TAA-injured rat livers. The expression of proliferation markers in TAA-injured rat livers (**a**,**b**) PCNA, (**c**,**d**) albumin. All experiments were repeated in duplicate to triplicate. * *p* < 0.05.

**Table 1 antioxidants-12-00046-t001:** List of primers used for qRT—PCR analysis.

Gene		Primers (Rat)	Tm (℃)	NCBI Ref.
**qRT-PCR**	PPARγ	F: 5’-GACAGACCTCAGGCAGATTG-3’ R: 5’-GTCAGCGACTGGGACTTTTC-3’	57	NM_013124.3
	Adiponectin	F: 5’-GACTGCCACTAATTCAGAGC-3’ R: 5’-CTCATGGGGATAACACTCAG-3’	55	NM_144744.3
	Adipsin	F: 5’-CACGTACCATGATGGGGCAA-3’ R: 5’-TCGAGATCCCACGTAACCA-3’	59	NM_000102.4
	LPL	F: 5’-ACAGGTGCAATTCCAAGGAAG-3’ R: 5’-CTTTCAGCCACTGTGCCATA-3’	57	M92059.1
	γH2AX	F: 5’-TGGAAAGGGTCAGGGAACG-3’ R: 5’-GACTTGTGCTGGTATCTGGGTG-3’	58	NM_001109291.1
	PINK	F: 5’-CATGGCTTTGGATGGAGAGT-3’ R: 5’-TGGGAGTTTGCTCTTCAAGG-3’	56	XM_032895606.1
	PARKIN	F: 5’-CTGGCAGTCATTCTGGACAC-3’ R:5’-CTCTCCACTCATCCGGTTTG-3’	57	XM_032894705.1
	8-OHdG	F: 5’-GGGCCCAAGCAGTGCTGTTC-3’ R: 5’-GATCCCTTTTTGCGCTTTTGC-3’	61	XM_034511266.1
	GAPDH	F: 5’-CGAGATCCCTCCAAAATCAA-3’ R: 5’-TGTGGTCATGAGTCCTTCCA-3’	55	NM_001357943.2
**TaqMan**	Mitochondrial D-loop	F: 5’-GGTTCTTACTTCAGGGCCATCA-4’ R: 5’-GATTAGACCCGTTACCATCGAGAT-3’	60	EU194676.1
	β-actin	F: 5’-GGGATGTTTGCTCCAACCAA-3’ R: 5’-GCGCTTTTGACTCAAGGATTTAA-3’	58	XM_039089807.1
	mitochondrial D-loop probe	JOE-TTGGTTCATCGTCCATACGTTCCCCTTA-3’		
	β-actin probe	FAM-CGGTCGCCTTCACCGTTCCAGTT-3’		

## Data Availability

The data presented in this study are available in the article.

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
