# Peer review of "Phosphatase of Regenerating Liver-1 (PRL-1)-Overexpressing Placenta-Derived Mesenchymal Stem Cells Enhance Antioxidant Effects via Peroxiredoxin 3 in TAA-Injured Rat Livers"

_antioxidants, 2022, doi:10.3390/antiox12010046_

Round 1

Reviewer 1 Report

Where did the authors get placenta-derived MSC? Which part of the perinatal tissue? The authors need to reformulate their nomenclature based on general consensus (Silini et al. doi.org/10.3389/fbioe.2020.610544). Have the authors tested human or rat PD-MSC?

Histological analyses offered are indeed quite limited (only H&E and TUNEL). What about the respiratory chain or expression of hepatic mitochondrial enzymes (such as CPS-1 and OTC)?

Can you confirm as necrotic cells (LDHpos cells) resulted higher in number in PD-MSC treated animals? After how many days were necrosis and apoptosis evaluated? When the analysis in Fig.1 have been performed? How many days post PD-MSC transplantation? What about the effects after 4 weeks? Any evidence of transplanted cells in short or long-term analysis? Since the animals were kept alive for 4 weeks, did you find any MSC after 4 wks?

IHC analysis for donor cells is required.

What about apoptotic/casp3pos cells+ any difference between treated and untreated animals?

In all figures: what is the difference between Nor and Naïve? replace “nor” with “healthy” or WT. (-) are intoxicated but untreated animals, aren’t they? Ntx (most likely non-transplanted animals) are the same as (-) aren’t they? Include legend and description in the Figure description

Line 204, the statement “DNA damage is caused by liver damage” is quite unclear and mostly incorrect. Please re-elaborate.

Affirmation in line 220 about the cirrhosis induction offered by sustained ox stress and TAA exposure requires additional information. After how many days does irreversible cirrhosis raise? What about fibrotic markers in early and late onsets?

ROS analysis shown in Figure 3: after how many days post-cell injection have been performed?

N values for every set of experiments need to be reported in Figure legends

Author Response

Reviewer #1:

Comments and suggestions for authors:

Point #1: Where did the authors get placenta-derived MSC? Which part of the perinatal tissue? The authors need to reformulate their nomenclature based on general consensus (Silini et al. doi.org/10.3389/fbioe.2020.610544). Have the authors tested human or rat PD-MSC?

  • Author’s response:

Thank you for your critical comments. As we mentioned in ‘Materials & Methods’, human placental samples were approved by the Institutional Review Board of CHA Gangnam Medical Center, Republic of Korea (IRB-07-18). According to our previous report, PD-MSCs were isolated from the chorionic plate of the human placenta (Lee et al., J Cell Biochem. 2010). As reviewer’s comments, I’ll revise the ‘Materials & Methods’ part to describe it in detail.

  • MJ Lee, J Jung, KH Na, JS Moon, HJ Lee, JH Kim, GI Kim, SW Kwon, SG Hwang and GJ Kim. Anti-fibrotic effect of chorionic plate-derived mesenchymal stem cells isolated from human placenta in a rat model of CCl(4)-injured liver: potential application to the treatment of hepatic diseases. J Cell Biochem. 2010;111(6):1453-63.

Point #2: Histological analyses offered are indeed quite limited (only H&E and TUNEL). What about the respiratory chain or expression of hepatic mitochondrial enzymes (such as CPS-1 and OTC)?

  • Author’s response:

We greatly appreciate the reviewer bringing up this important point. We also agree that further analysis of hepatic mitochondrial enzymes is necessary to convincing the results. Unfortunately, we consider the 10-day period to be limited to purchasing and analyzing antibodies related to hepatic mitochondrial enzyme. Therefore, we will analyze the hepatic mitochondrial enzymes as you advised through further study.

Point #3: Can you confirm as necrotic cells (LDHpos cells) resulted higher in number in PD-MSC treated animals? After how many days were necrosis and apoptosis evaluated? When the analysis in Fig.1 have been performed? How many days post PD-MSC transplantation? What about the effects after 4 weeks? Any evidence of transplanted cells in short or long-term analysis? Since the animals were kept alive for 4 weeks, did you find any MSC after 4 weeks?

  • Author’s response:

Thank you for your critical comments. We observed that the increased number of necrotic cells in thioacetamide-induced animal models decreased after transplantation of Naïve PD-MSCs and PRL-1 overexpressed PD-MSCs. We analyzed the necrotic cells by TUNEL assay in liver samples collected 4 weeks after transplantation of PD-MSCs. I would like to proceed with the analysis of necrotic cells after 4 weeks by referring to your comment, but there is a limit to meeting the revision time of 10 days for animal experiments over 19 weeks.  We’ll conducted the long-term analysis after transplantation in a further study. Also, evidence of transplantation of PD-MSCs was reported in our previous report analyzing PKH67-labed PD-MSCs and hAlu sequences (Seok et al, Antioxidant. 2020). According our previous reports, we confirmed that PD-MSCs were only observed in 1-2 weeks after transplantation and their therapeutic efficacy appeared even after 4 weeks.

  • Seok J, Park H, Choi JH, Lim JY, Kim KG, Kim GJ. Placenta-Derived Mesenchymal Stem Cells Restore the Ovary Function in an Ovariectomized Rat Model via an Antioxidant Effect. Antioxidants (Basel). 2020 Jul 6;9(7):591.

Point #4: IHC analysis for donor cells is required.

  • Author’s response:

Thank you for your critical comments. Referring to your comments, it is important to do IHC analysis on donor cells. However, in order to perform IHC analysis on donor cells, an antibody that can be used as a positive control must be ordered and analyzed. There is a limit to analyzing in 10 days.

Point #5: What about apoptotic/casp3pos cells+ any difference between treated and untreated animals?

  • Author’s response:

Thank you for your critical comments. Although hepatocyte occupies a large part in liver tissue, we also pooled individual and confirmed that there are many cells in the liver at the gene level as a homogenized protein. Cleaved-caspase 3 was significantly increased in the Normal group compared that in the NTx group. In addition, in the groups transplanted with stem cells, no significant results were obtained compared to the non-transplanted group. It would be good if other genes related to apoptosis were identified and analyzed clearly, but this also has limitations in antibody stock, so apoptosis was further analyzed at tissue staining and serum levels. As a result, as shown in TUNEL assay, apoptotic expression increased in the TAA-induced group, but decreased expression in the stem cell transplanted group. When analyzed through LDH assay at the serum level, the following results were obtained. So, in conclusion, we were able to confirm that Naïve PD-MSCs and PRL-1 overexpressing PD-MSCs have anti-apoptotic effects because we observed a decrease in apoptosis in the stem cell transplanted group through serum and tissue staining.

Point #6: In all figures: what is the difference between Nor and Naïve? replace “nor” with “healthy” or WT. (-) are intoxicated but untreated animals, aren’t they? Ntx (most likely non-transplanted animals) are the same as (-) aren’t they? Include legend and description in the Figure description

  • Author’s response:

Thank you for your critical comments. However, it is known that the wild type notation is used in groups that are not genetically mutated in nature. Since we are a chemical-induced model using TAA and not a model that induces genetic modification, we can use the term normal.

Point #7: Line 204, the statement “DNA damage is caused by liver damage” is quite unclear and mostly incorrect. Please re-elaborate.

  • Author’s response:

Thank you for your critical comments. As reviewer’s comments, we added the detail into the revised manuscripts.

Point #8: Affirmation in line 220 about the cirrhosis induction offered by sustained ox stress and TAA exposure requires additional information. After how many days does irreversible cirrhosis raise? What about fibrotic markers in early and late onsets?

  • Author’s response:

Thank you for your critical comments. We conducted experiments based on previous studies. In the study, the concentration of TAA and the duration of the study were discussed in detail. According to studies, hepatitis and hepatic fibrosis are induced at least 6-8 weeks, and liver cirrhosis occurs when induced at least 12 weeks. Furthermore, when induced for more than 50weeks, cells dysplasia becomes evident and malignant tumors are formed, but HCC is not formed.

  • Wallace M, Hamesch K, Lunova M, et al. Standard Operating Procedures in Experimental Liver Research: Thioacetamide model in mice and rats. Laboratory Animals. 2015;49(1_suppl):21-29.
  • Deng X, Zhang X, Li W, Feng RX, Li L, Yi GR, Zhang XN, Yin C, Yu HY, Zhang JP, Lu B, Hui L, Xie WF. Chronic Liver Injury Induces Conversion of Biliary Epithelial Cells into Hepatocytes. Cell Stem Cell. 2018 Jul 5;23(1):114-122.e3.

Point #9: ROS analysis shown in Figure 3: after how many days post-cell injection have been performed?

  • Author’s response:

Thank you for your critical comments. In Figure.3, we analyzed the ROS levels by MitoSOX-MitoTracker staining in liver samples collected 4 weeks after transplantation of PD-MSCs.

Point #10: N values for every set of experiments need to be reported in Figure legends.

  • Author’s response:

Thank you for your critical comments. As reviewer’s comments, we added the detail into the revised manuscripts.

Reviewer 2 Report

The manuscript entitled “Phosphatase of regenerating liver-1 (PRL-1)-over expressing placenta-derived mesenchymal stem cells enhance antioxidant effects via peroxiredoxin 3 in TAA-injured rat livers” submitted by Hee Jung Park et al. is an excellent evaluation of the role of PRL-1  in PD-MSCs in promoting antioxidant activity in chemically injured rat livers. This manuscript is through and comprehensive evaluation of the effects as well as the key players involved in this process. By in large the manuscript is an excellent story and only very minor corrections are included that may aid in the ease of reading. Great job!

Minor corrections:

-Modest wording clarification: Line 176 would be potentially better phrased as: “To induce a cirrhosis model, rats received intraperitoneal injections of TAA (300mg/kg) twice a week for 12 weeks”

- It may be worth including the details of the cell injection in line 177 (although covered in the intro, not so much in the results). Just a line saying that one set received naïve cells and the other PRL-1+. As of now it just is assumed

- A sentence explaining what the TUNEL assay detects would be of great value.

- Figure 1: The H&Es are labeled as “f” currently. No clear explanation of what the yellow arrows are indicating.

Author Response

Reviewer #2:

Comments and suggestions for authors:

The manuscript entitled “Phosphatase of regenerating liver-1 (PRL-1)-over expressing placenta-derived mesenchymal stem cells enhance antioxidant effects via peroxiredoxin 3 in TAA-injured rat livers” submitted by Hee Jung Park et al. is an excellent evaluation of the role of PRL-1 in PD-MSCs in promoting antioxidant activity in chemically injured rat livers. This manuscript is through and comprehensive evaluation of the effects as well as the key players involved in this process. By in large the manuscript is an excellent story and only very minor corrections are included that may aid in the ease of reading. Great job!

Author’s reply:

We greatly appreciate the reviewer’s positive statement that “Phosphatase of regenerating liver-1 (PRL-1)-over expressing placenta-derived mesenchymal stem cells enhance antioxidant effects via peroxiredoxin 3 in TAA-injured rat livers”. We modified some error word as well as reviewer’s critical comments.

Point #1: Modest wording clarification: Line 176 would be potentially better phrased as: “To induce a cirrhosis model, rats received intraperitoneal injections of TAA (300mg/kg) twice a week for 12 weeks”

  • Author’s response:

Thank you for your critical comments. As reviewer’s comments, we revised the better wording in manuscripts.

Point #2: It may be worth including the details of the cell injection in line 177 (although covered in the intro, not so much in the results). Just a line saying that one set received naïve cells and the other PRL-1+. As of now it just is assumed

  • Author’s response:

We greatly appreciate the reviewer bringing up this important point. Referring to your comments, we added the detail into the revised manuscripts.

Point #3: A sentence explaining what the TUNEL assay detects would be of great value.

  • Author’s response:

We greatly appreciate the reviewer bringing up this important point. Referring to your comments, we added the detail the detection of necrotic cells by TUNEL assay into the revised manuscripts.

Point #4: Figure 1: The H&Es are labeled as “f” currently. No clear explanation of what the yellow arrows are indicating.

  • Author’s response:

Thank you for your critical comments. As reviewer’s comments, we added the detail into the Figure.1 legends of revised manuscripts.